# Freezing and piercing of in vitro asymmetric plasma membrane by α-synuclein

Paul Heo[1✉] & Frederic Pincet [1✉]

Synucleinopathies are neurological diseases that are characterized by the accumulation of aggregates of a cytosolic protein, α-synuclein, at the plasma membrane. Even though the pathological role of the protein is established, the mechanism by which it damages neurons remains unclear due to the difficulty to correctly mimic the plasma membrane in vitro. Using a microfluidic setup in which the composition of the plasma membrane, including the asymmetry of the two leaflets, is recapitulated, we demonstrate a triple action of α-synuclein on the membrane. First, it changes membrane topology by inducing pores of discrete sizes, likely nucleated from membrane-bound proteins and subsequently enlarged by proteins in solution. Second, protein binding to the cytosolic leaflet increases the membrane capacitance by thinning it and/or changing its relative permittivity. Third, α-synuclein insertion inside the membrane hydrophobic core immobilizes the lipids in both leaflets, including the opposing protein-free extracellular one.

[1] Laboratoire de Physique de l'Ecole normale supérieure, ENS, Université PSL, CNRS, Sorbonne Université, Université de Paris, F-75005 Paris, France.
✉email: paul.heo@ens.fr; frederic.pincet@ens.fr

α-synuclein (αSyn) is a cytosolic protein that has been associated for decades with the pathogenesis of neuro-pathologies known as synucleinopathies, such as Parkinson's disease, dementia with Lewy bodies, and multiple system atrophy[1–4]. Native αSyn is an essentially unstructured protein in the cytosol[5–7]. It is well established that the starting point of αSyn pathological action is its accumulation, probably as α-helices, on negatively charged membranes such as the plasma membrane[2,8–13]. However, how this accumulation disrupts neuronal function and leads to neuronal death remains largely unclear[2,14–18]. Recent in vitro studies proposed that αSyn may be able to form pores[4,19–24] in membranes which would be lethal to neurons. But applying their hypothesis to physiology may not be appropriate since all the model membranes that were used have a lipid composition very different from that of the plasma membrane. One notable peculiarity of the plasma membrane is that it is very asymmetric: the cytosolic and extracellular leaflets have completely different compositions[25–29]. Even though this asymmetry is required for physiological membrane processes, it is difficult to reproduce on model membranes.

We recently developed a microfluidic setup allowing the formation of horizontal lipid membranes with a controlled composition of each leaflet[30]. Furthermore, the electrical and optical properties of the membranes can be simultaneously measured. Hence, this setup is perfectly adapted to test the pore hypothesis on an asymmetric membrane. We find that αSyn induces pores that are more stable with asymmetric membranes than with symmetric membranes. The pores display discrete perimeters that are consistent with the addition of unit-blocks on the latest expanded pore. The unit-block, which is likely to contain either one protein that spans the membrane thickness twice or two proteins, each spanning it once only, increases the perimeter of the pore by 2–3 nm. Varying the αSyn concentration in the cytosolic side, we suggest that the initial pore is nucleated from αSyn bound to the membrane and contains two unit-blocks, and the subsequent unit-blocks seem to come one at a time from the αSyn remaining in solution. We also took advantage of our setup to visualize the membrane and found, using fluorescence recovery after photobleaching (FRAP), that the presence of membrane-bound αSyn reduces the membrane fluidity. Surprisingly, when αSyn is bound to one leaflet, the fluidity of the other leaflet is also impaired. Finally, αSyn increases the overall membrane capacitance. This increase can only be due to a decrease in membrane thickness and/or change in relative permittivity. These results show that binding of αSyn to the membrane simultaneously induces changes in the membrane topology, morphology, and fluidity which prevent normal behavior of neurons and eventually neuronal death.

## Results

**αSyn disrupts asymmetric membranes**. All experiments presented here were performed using a microfluidic setup we recently developed[30]. In a poly(dimethylsiloxane) (PDMS) chip, two horizontally crossed-channels are vertically separated by a tiny cylindrical hole (Fig. 1a and Methods). The lower channel is directly above a microscope coverslip; the upper one is right above the cylindrical hole. The chip is initially filled with squalene. To form asymmetric membranes having plasma membrane-like composition[26], two sets of small unilamellar vesicles (SUVs) were prepared in a 150 mM KCl, 25 mM HEPES, pH 7.4 buffer and flowed in each channel (Fig. 1b). The first (resp. second) set had a lipid composition resembling that of the cytosolic (resp. extracellular) leaflet of the plasma membrane (see Methods for detailed compositions) and was flowed in the upper/cytosolic (resp. lower/extracellular) channel, trapping a 1 nl squalene

droplet in the cylindrical hole. The SUVs spontaneously fused with the squalene droplet to form a monolayer at each squalene/buffer interface as previously described[30,31]. A key to the functionality of this setup is the complete absorption of the squalene droplet by PDMS. During this absorption process, the two monolayers meet and zip over ~30 min to form an oil-free asymmetric membrane[30]. Membrane formation is simultaneously monitored by bright-field or fluorescence microscopy and capacitance measurements (Fig. 1c, d). Without αSyn, the bilayer is extremely stable for ~3 h after complete zipping. When αSyn was flowed in the cytosolic channel, the membrane became covered with αSyn in a few minutes (Fig. 1e). The accumulation of αSyn was followed by rapid rupture of the membrane (Fig. 1b and f). In the range of concentrations that we tested, 5 nM to 250 nM, the lifetime was reduced to ~15 min on average (Fig. 1g). Hence the presence of αSyn accelerates membrane rupture. Besides, the mean lifetime does not seem to depend on the concentration (Fig. 1g), suggesting that the process of αSyn binding to the membrane is fast enough to be completed much before membrane rupture.

As a comparison, we added αSyn (25 nM) in the extracellular channel and observed that the lifetime was reduced to a lower extent than when αSyn is in the cytosolic channel, ~45 min instead of ~15 min (Supplementary Fig. 1a).

Finally, we tested symmetric membranes with a lipid composition resembling the cytosolic leaflet (see Methods for detailed composition). Titrating αSyn in one channel, the lifetime reduced much less than in the case of the asymmetric membrane: it varied from almost 3 h at 5 nM down to ~30 min at 250 nM (Fig. 1g). We repeated this titration while also having αSyn in the other channel at 60 nM. The result was an additional decrease in the membrane lifetime that became similar to the one observed with the asymmetric membrane when there was 75 nM or more αSyn in the other channel (Supplementary Fig. 1b).

**αSyn forms long-lasting pores in asymmetric membranes**. It was previously reported that αSyn induces pores, i.e., "holes" in the membrane. To test whether such pores exist with asymmetric membranes and could cause membrane rupture, we performed electrical current measurements. After αSyn was flowed in the cytosolic channel during monolayer zipping, a 100 mV constant voltage was applied between the two channels (Fig. 2a). Two αSyn concentrations were tested, 25 and 250 nM. Initially, no current was detected, showing that there was no aqueous connection between the two sides of the membrane (Fig. 2a–c). About 15 min after flowing the αSyn, an increase in current occurred, indicating the opening of a pore in the membrane. With 250 nM αSyn, the current continued to steadily increase and reached an out of range value within 10 ms. At that point, the membrane was broken, as confirmed by fluorescence (Fig. 1f). When 25 nM αSyn was used, the kinetics of the pore was slower with a step-wise variation of the current. Overall the current was increasing, but it sometimes transiently decreased. As in the 250 nM case, the membrane eventually ruptured after pore opening but the pore lifetime was drastically changed: it lasted longer, between 10 and 30 s. Hence, decreasing 10 times the bulk αSyn concentration did not affect the delay before pore formation but increased the pore lifetime more than 100 times.

The presence of pores was also tested with symmetric membranes at 250 nM αSyn. As expected, pores were observed (Fig. 2d). However, the pore kinetics was strikingly different from that observed with asymmetric membranes. Pores were able to reseal as attested by pore opening followed by the disappearance of any current. This is consistent with previous observations[4,19,24]. Hence, several pores could sequentially be formed on the same

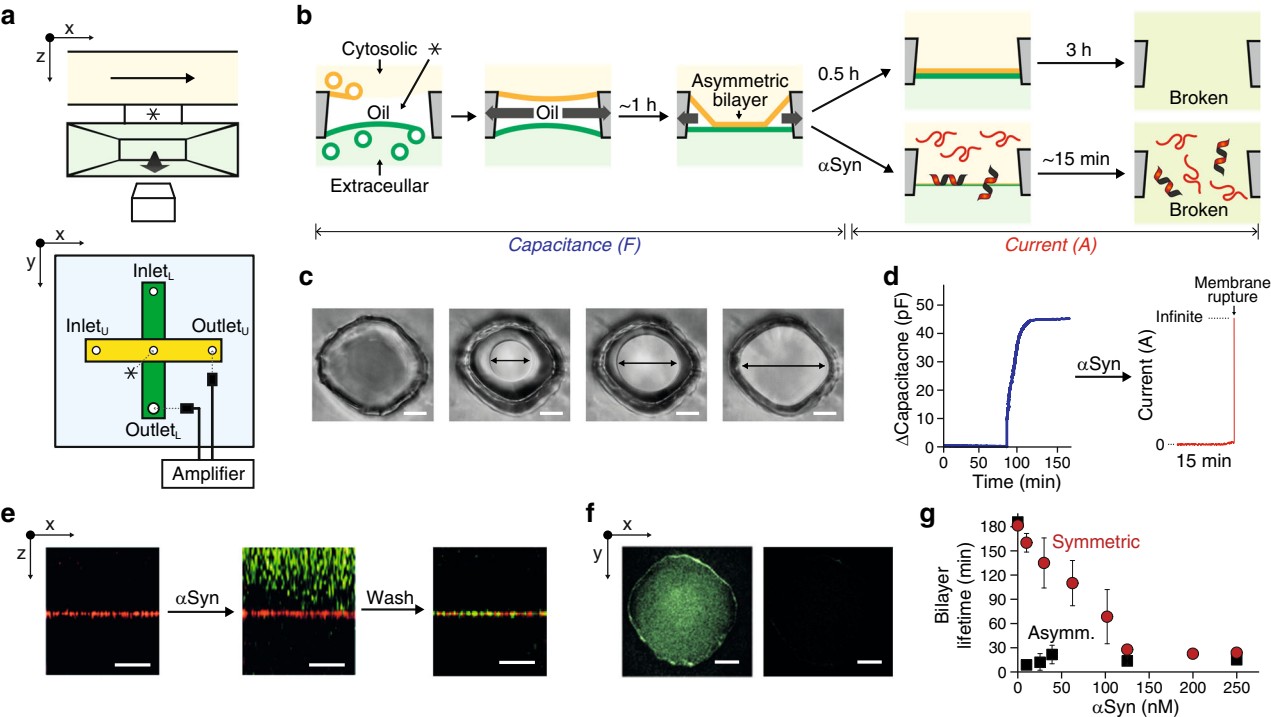

**Fig. 1 Probing αSyn action on asymmetric plasma-like membranes. a** Sketch of the microfluidic setup. A PDMS chip contains two horizontally crossed-channels that are vertically separated by a tiny cylindrical hole indicated by an asterisk, * (also in (**b**)). The chip is mounted on a microscope coverslip. The upper channel is colored in yellow, the lower one in green. An electrode is inserted in each channel. Optical and electrical observations can be performed simultaneously. **b** Schemes of αSyn-mediated membrane lifetime assay. Each extracellular and cytosolic monolayer is formed by (i) pushing away oil (squalene), and (ii) spontaneous fusion of SUVs with the corresponding lipid composition at each oil/buffer interface. Upon absorption of the oil by PDMS, the two monolayers zip and form an asymmetric bilayer without any trace of residual squalene[30]. The bilayer remains stable for typically 3 h. When αSyn is added, the lifetime is reduced to ~15 minutes. After almost complete zipping of the membrane, the system was switched to "current mode" by applying 100 mV between the two channels. **c** Bright-field pictures of the bilayer zipping process. Arrows indicate the bilayer diameter. Scale bar, 25 μm. **d** Time-dependent increase of the capacitance between the two channels during bilayer zipping followed by a current trace after switching to current mode with 100 nM αSyn (red). The final large current increase is due to membrane rupture. **e** x-z sections of z-stack images performed after bilayer formation, after flowing αSyn, and after washing of αSyn. SUVs contained 0.2% Atto 647N-DOPE and αSyn was labeled with Atto488. The membrane is the red horizontal line. αSyn (green), binds and accumulates on the membrane. **f** Fluorescence image at the membrane (x-y plane) before (top) and after (bottom) membrane rupture. Green indicates Atto488 labeled αSyn. **g** Comparison of asymmetric (black) and symmetric (red) bilayer lifetime with αSyn concentration in the cytosolic channel. Error bars are standard errors of the mean. Lipid composition of asymmetric membrane in (**c**) to (**g**) is DOPC:DOPS:Cholesterol:DOPE:PIP$_2$ at 7:15:45:30:3% and DOPC:DOPS:Cholesterol:DOPE:Sphingomyelin at 20:5:45:15:15% in mol, for the inner and extracellular leaflet mimic respectively. The symmetric membrane is made of DOPC:DOPS:Cholesterol:DOPE:PIP$_2$ at 7:15:45:30:3% in mol.

membrane without disrupting it. Some pores opened with a specific current value and resealed without any step-increase in intensity while others displayed a few current step-increases prior to disappearing. The lifetime of these pores was typically 10–100 ms. In some cases, a cascade of transient pores occurred in 100 ms or less. Because of the low probability to form two independent pores in such a short time, we attribute this type of behavior to the same proteins that remain correctly positioned to form a pore, i.e., the same pore cycles between an open and closed conformation (Fig. 2d). Eventually, the membrane broke while a pore was open, typically ~30 min after αSyn was flowed, and ~4 pores were formed during membrane lifetime.

We measured pores with similar kinetics during bilayer zipping and with fully expanded bilayers. This similarity rules out any artifact that may come from the interaction between αSyn, the membrane and the PDMS chip.

**The rim of pores is made of unit-blocks with 2 α-helices.** For any given current trace, discrete distributions of the current intensity, $I$, are observed (Fig. 3a and Supplementary Fig. 2). Two non-exclusive processes can be envisioned to explain these discrete distributions: (i) the successive opening and closing of

several identical pores, or (ii) the enlargement of a single pore. To differentiate between these two possibilities, we analyzed 193 current jumps between two steps and evaluated the current step intensity value, $I_n$, where $n$ is the number of the step. The $n^{th}$ jump, as seen in Supplementary Fig. 2, is $I_n - I_{n-1}$. If identical pores sequentially opened, $I(n)$ should be a multiple of the intensity of a single pore, $I_1$, and therefore should vary as $nI_1$. This linear variation is not observed experimentally (Fig. 3b), showing that sequential opening of multiple pores is not happening. Thus, the existence of a single pore of various sizes seems more likely. The pore diameter variation supports this hypothesis. Indeed, for a cylindrical pore, $I$ is related to the pore diameter, $d$, and length, $l$, through:

$$I = \frac{\pi GCUd^2}{4l} \qquad (1)$$

where $G$ is the molar conductivity (~10 S M$^{-1}$m$^{-1}$), $C$ the concentration (~150 mM) and $U$ the voltage (100 mV). Using this approximation, the diameter of the $n^{th}$ step, $d(n)$, varies linearly with $n$ (Fig. 3c). Equivalently, the perimeter of the pore, i.e., the length of the rim of the pore, $p(n)$, also varies linearly with $n$. This linear dependency shows that, for each step, a constant length is

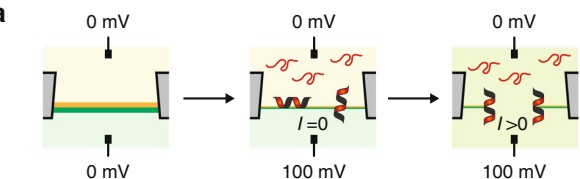

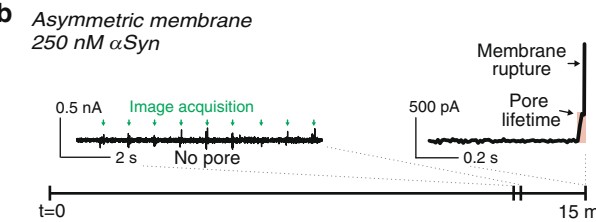

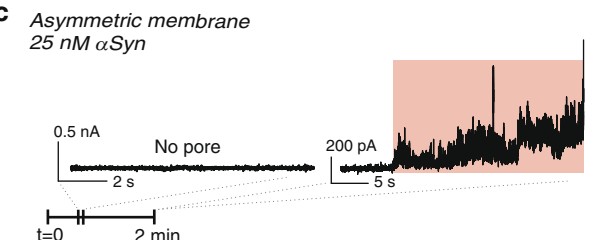

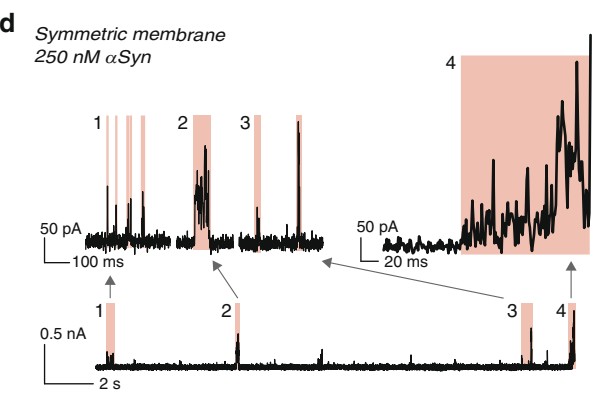

**Fig. 2 αSyn pore formation on asymmetric and symmetric membranes.** **a** Schematic illustration of αSyn pore formation and monitoring. After membrane formation, αSyn is injected into the cytosolic channel and 100 mV is applied from the cytosolic to the extracellular face of the membrane. Upon pore opening, a non-zero current is measured. **b**, **c** Pore opening and asymmetric membrane rupture with 250 nM (**b**) or 25 nM (**c**) αSyn. Initially, there is no current, confirming the membrane sealed the hole separating the channels. After a few minutes, an increase in current is observed, showing a pore has opened between the channels (highlighted in red). Then, the membrane is broken as attested by out-of-range current (>10 nA). The little glitches (green arrows) observed in the "no pore" section of the current trace correspond to the image acquisition by the camera. **d** Pore opening with symmetric membranes. Current traces show that transient pores open and reseal frequently until the membrane is broken. **a**–**d** The lipid compositions are the same as in Fig. 1.

added to the perimeter, suggesting that the pore is expanded by inserting a nano-size unit-block of αSyn. This is consistent with another study on symmetric membranes in which three sequential steps were observed[24]. This unit-block may contain either one protein or several proteins that simultaneously bind at the rim.

Strikingly, the variation of $d$ with $n$ intercepts the $x$-axis at $n = -1 \pm 0.5$. This indicates that the first step contains 2 unit-blocks. Because the edge of the pore is probably made of α-helices, three or more α-helices must span the membrane to actually have a hole in

the center. Since a newly nucleated pore has two unit-blocks, there must be four α-helices or more in the initial pore edge. To estimate the dimension of the unit-block, $p/n$, from Eq. 2, a value must be chosen to the pore length. It is larger than the membrane thickness, 5 nm. In addition, two α-helices were found in micelle-bound αSyn[32]. It is difficult to assume that the pore can exceed their length, i.e., ~40 residues or, equivalently, ~10 nm. Assuming the thickness of the pore is 5 nm (resp. 10 nm), the perimeter would increase by 2.2 ± 0.8 nm (resp. 3.1 ± 1 nm) per unit-block (Fig. 3d). The diameter of an α-helix is typically 1.2 nm. Hence, a 2–3 nm increase corresponds to the addition of two α-helices. With this analysis, the initial pore is made of four α-helices and each subsequent step-increase in current corresponds to the addition of two α-helices. Thus, each unit-block contains either one protein spanning twice the membrane or two proteins spanning the membrane once (Fig. 3e).

Finally, because pore formation time seems independent of αSyn concentration, pore nucleation is likely due to proteins that are already membrane-bound. However, the pore kinetics was slower for lower bulk αSyn concentration, indicating that the subsequent unit-blocks come from proteins in solution and not from the membrane. Since oligomeric states of αSyn do not seem to exist in solution[33], we favor a model in which αSyn is added one at a time and each protein contributes to two additional α-helices. Then, the initial pore contains two proteins.

There was no quantitative difference between the asymmetric and the symmetric membranes regarding the values of the step-intensities of the current, suggesting the pore structure is independent of the membrane composition.

**αSyn increases the membrane capacitance.** Binding and accumulation of αSyn on membranes induces pore formation and membrane rupture. Hence, there are strong interactions between αSyn and the lipid in the cytosolic leaflet and likely part of the extracellular leaflet that lead to topological changes. How these interactions affect the lipid arrangement is unclear. To address this question, we have measured the capacitance of the membrane during αSyn binding. In the presence of 100 nM αSyn in the cytosolic channel, the capacitance was 20% larger than without protein after full zipping of the bilayer (Fig. 4a and Supplementary Fig. 3). This was not observed with a control transmembrane protein for which the final capacitance actually decreased (t-SNARE, target membrane-soluble N-ethylmaleimide-sensitive factor attachment protein receptor). The membrane capacitance, $C_m$, is related to the hydrophobic thickness, $e$, and area, $A$, of the bilayer through:

$$C_m = \frac{\varepsilon_0 \varepsilon_r}{e} A \qquad (2)$$

Thus, an increase in capacitance is likely due to an increase in area, a decrease in thickness or a change in relative permittivity, $\varepsilon_r$. Since lipids are incompressible, the total volume of a lipid molecule remains constant which implies that a decrease in membrane thickness is automatically associated with a commensurate increase in membrane area. Hence, the increase in capacitance can only have two non-mutually exclusive origins: simultaneous thinning and area expansion of the membrane and/or increase in relative permittivity. In the first case scenario, a ~20% change in capacitance could correspond to a simultaneous ~10% increase of the membrane area and ~10% decrease of the membrane thickness. This thinning/stretching of the membrane may originate from the insertion of αSyn that force the lipid leaflet to adopt a thinner conformation, at least locally (Fig. 5). A 20% increase in relative permittivity of $\varepsilon_r$ would be necessary to account for the capacitance increase. However, since thinning/

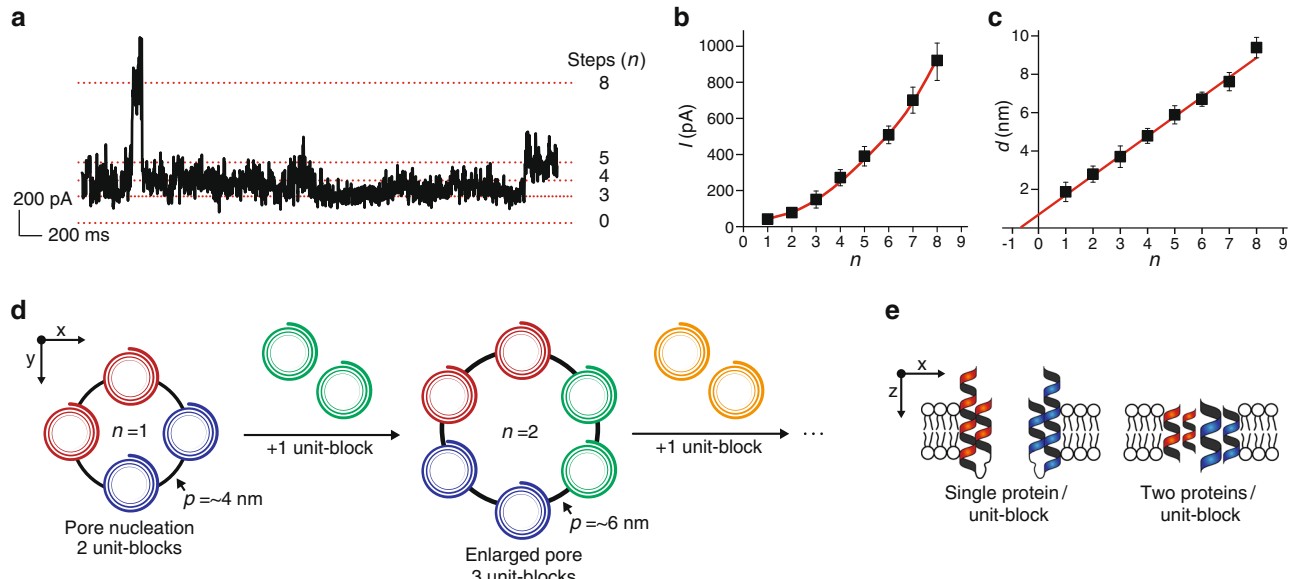

**Fig. 3 Pore nucleation and rim expansion defined by discrete unit-blocks of αSyn. a** Current trace displaying multiple current steps. **b** Non-linear variation of the current intensity with the step number. Statistics were obtained on 23, 24, 35, 27, 22, 30, 25 and 7 events for step number 1, 2, 3, 4, 5, 6, 7 and 8 respectively. **c** The diameter of a single pore as deduced from Eq. 1 varies linearly with the number of the step and the intercept of the x-axis is near −1. In (**b** and **c**), error bars are standard deviations. **d** Sketch of the αSyn pore formation defined by unit-blocks. The initial pore (black circle) contains two unit-blocks (red and blue) corresponding to four α-helices. A first additional unit-block (green) adds two α-helices to the initial pore and other extra unit-blocks (yellow) can be subsequently added in a similar manner. The pore rim length increase is 2–3 nm per unit-block, which is consistent with two α-helices. **e** Side view sketch of initial pore defined by the number of αSyn. The α-helices of a unit-block come either from a single αSyn or from two αSyn, each contributing to one α-helix.

stretching of the membrane by αSyn binding was previously reported[34], we favor this explanation without excluding a possible increase in relative permittivity. In any case, to increase the membrane capacitance, αSyn must interact deep inside the hydrophobic core of the membrane which may affect membrane fluidity; this is what we tested next.

**αSyn on the cytosolic leaflet freezes the other leaflet.** FRAP was used to evaluate the effect of αSyn on the membrane fluidity (Fig. 4b). This was done on symmetric membranes because the working time was too short on asymmetric ones before rupture in the presence of αSyn. As expected, without protein, both leaflets of the membrane were completely fluid (Supplementary Fig. 4)[35]. Upon adding αSyn in one channel at 100 nM, 50% of the lipids in the cis-leaflet were immobile, confirming the strong interaction between the proteins and the lipids. Surprisingly, the trans-leaflet was also affected: 25% of the lipids in the αSyn-free leaflet were immobile. This shows unambiguously that αSyn penetrates deep inside the membrane and interacts with the trans-lipids, immobilizing 25% of them. When αSyn is added in both channels, 75% of the lipids are immobilized. Using fluorescent αSyn showed that the membrane-bound proteins are completely immobile, as expected[36,37]. Hence, αSyn binds membrane tightly which freezes the extracellular leaflet.

## Discussion
Overall, these results indicate that αSyn takes nanometric control of the membrane topology, morphology and fluidity. We propose a putative molecular model of the in vivo process that leads to neuronal malfunction and eventually cellular death (Fig. 5a, b). Initially, αSyn is monomeric in the cytosol. It quickly binds to the cytosolic leaflet. We cannot differentiate between a lateral loading (αSyn laying flat on the cytosolic leaflet) and longitudinal insertion (αSyn crosses the membrane). However, since αSyn interacts

with both leaflets of the membrane, it is likely that a fraction of the αSyn is longitudinally inserted. As a result, αSyn immobilizes a large fraction of the lipids, which induces a defective fluidity that freezes the membrane. Finally, because αSyn penetrates deep inside the membrane, it occasionally achieves a molecular arrangement comprising four α-helices emanating from two proteins that leads to the nucleation of a pore. Once the pore is open, it does not reseal for asymmetric plasma membranes. Then, αSyn molecules from the solution bind at the rim one at a time and enlarge the pore, each protein adding two α-helices to the pore perimeter.

Pore expansion is highly dynamic: αSyn from the solution can bind and unbind. The pores are more stables in the case of physiological membranes compared to symmetric ones. Quantitatively, this difference in stability is due to different binding and unbinding rates, which can be roughly estimated. Typically, a new unit-block is added every 10 ms at 25 nM αSyn with the asymmetric membrane and at 250 nM αSyn with the symmetric membrane. The resulting binding rates are therefore of the order of 4 s$^{-1}$nM$^{-1}$ for the asymmetric bilayer and 0.4 s$^{-1}$nM$^{-1}$ for the symmetric bilayer. The unbinding rate is about 100 s$^{-1}$ for both asymmetric and symmetric bilayers. Hence, the corresponding affinity would be ~10 times better for the asymmetric membrane: 25 nM vs. 250 nM. These affinity values were to be expected as long-lasting pores were observed at 25 nM for the asymmetric membrane and 250 nM for the symmetric one. The affinity difference between the two types of membranes is probably due to a lower line tension of the rim. The composition of the asymmetric membrane likely favors high local curvatures and/or a better insertion of the protein because αSyn is sensitive to hydrophobicity, or equivalently, curvature[38]. This difference is also consistent with the decrease in membrane lifetime (Fig. 1g). The affinity values remain in the range of previously published affinities for the binding of αSyn with membranes, 65 nM[39]. This quantitative analysis is valid only for the kinetics after pore

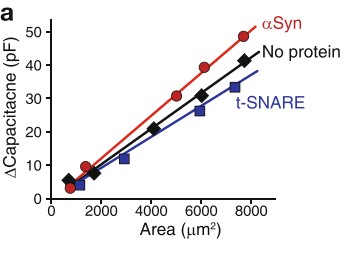

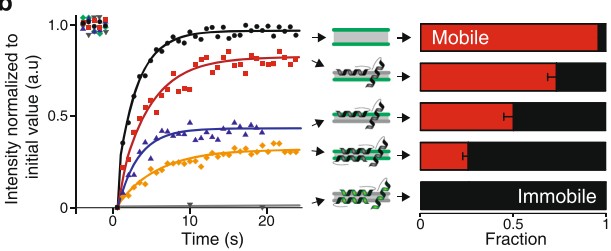

**Fig. 4 Capacitance increase and freezing of the membrane upon binding of αSyn. a** Capacitance alteration by membrane proteins. The specific capacitance is the slope of the linear dependence of the capacitance with the membrane area during bilayer zipping without any protein (black), with 100 nM αSyn (red) and t-SNARE (blue). **b** Fluorescence recovery after photobleaching of a 10 μm disk when monitoring: (Top, black) the lipids of both leaflets without any αSyn; (Second to the top, red) the lipids of the extracellular leaflet with 100 nM αSyn in the cytosolic channel; (middle, blue) the lipids of the cytosolic leaflet with 100 nM αSyn in the cytosolic channel; (fourth to the top, yellow) the lipids of both leaflets with 100 nM αSyn in both channels; (bottom, grey) labeled αSyn when flowed in at 100 nM in both channels. The mobile and immobile fractions in each case are displayed in red and black respectively. Error bars are standard errors of the mean. The diffusion coefficient of the mobile fraction is typically 12 μm²/s. Green in the cartoon (middle column) indicates the fluorescent leaflet(s) or αSyn. The lipid composition is the same as in Fig. 1.

nucleation and is not valid for the nucleation or complete resealing of the pore which entails a different molecular process.

The physiological consequences of these membrane modifications are dramatic and potentially lethal to the cell (Fig. 5c). A putative decrease in membrane thickness may alter the insertion of some transmembrane proteins because of the mismatch between the hydrophobic core of the bilayer and the length of the trans-membrane domain[40,41]. This mismatch may lead to incorrect folding and functionality of the protein. The reduced fluidity of the membrane prevents protein diffusion and, consequently, most physiological processes that normally occur at the plasma membrane. For instance, it would prevent or slow down the formation of molecular complexes or membrane fusion[42–45]. Finally, the formation of stable pores disrupts the molecular gradients between the cytosolic and extracellular media[46–48]. An example is calcium gradient: calcium concentration is actively maintained at a low level in the cytosol (in the 1 μM range or less) and high level outside of the cell (~1 mM). The equilibration between the inner and outer calcium concentration suppresses any evoked-neuronal activity. Also, soluble αSyn could pass through the pore and start accumulating on the extracellular leaflet of the plasma membrane. The presence of extracellular αSyn was actually observed in vivo. It may enhance the immobilization of membrane-bound molecules and facilitate membrane disruption.

## Methods

**Microfluidic chip fabrication.** The bilayer-forming chip was constructed as previously described[30]. In brief, a Titan2 HR (Kudo3D) 3D-printer was used to provide the two grooves (10 mm × 400 μm × 200 μm and 5 mm × 400 μm × 400 μm, length × width × height) for the channels (upper and lower, respectively) and the

cylindrical hole (diameter ~120 μm × length ~100 μm) for the bilayer formation. After assembling the two molds, a 9:1 mixture of Sylgard 184 (Dow Corning) was poured, cured at 72 °C for 50 min, and detached for subsequent curing at 72 °C for 12 h. Additional PDMS cubes, 10 mm × 3 mm × 5 mm (length × width × height) punched with 0.5 mm Biopsy Punch (World Precision Instruments) and 3 mm × 3 mm × 5 mm (length × width × height) PDMS, were glued with PDMS mixture to create top cover and bottom tube holder, respectively. After punching the bottom tube holder with 0.5 mm Biopsy Punch, the lower channel was created by sealing the bottom groove with a Precision Cover Glass #1.5 (Thorlabs) by plasma cleaner (Harrick Plasma). After cleaning with acetone and isopropanol for 5 min with sonication and drying at 24 °C for 2 h, the chips which absorbed the 1 nl trapped oil during bilayer formation in ~1 h were selected for subsequent experiments.

**Optical and electrical monitoring of membrane formation.** Two lipid monolayers on each side of squalene droplet were separately formed by different SUVs composition: DOPC:DOPS:Cholesterol:DOPE:PIP₂ at 7:15:45:30:3% in mol for the monolayer adjacent to cytosolic (upper) channel, DOPC:DOPS:Cholesterol:DOPE:Sphingomyelin at 20:5:45:15:15% in mol for the other adjacent to extracellular (bottom) channel. In the case of symmetric bilayer, the extracellular leaflet had the same composition as the cytosolic one: DOPC:DOPS:Cholesterol:DOPE:PIP₂ at 7:15:45:30:3%. The abbreviations of the lipids are: DOPC, 1,2-dioleoyl-sn-glycero-3-phosphocholine; DOPS, 1,2-dioleoyl-sn-glycero-3-phosphoserine; DOPE, 1,2-dioleoyl-sn-glycero-3-phosphoethanolamine; PIP₂, 1,2-dioleoyl-sn-glycero-3-phospho-(1'-myo-inositol-4',5'-bisphosphate) (ammonium salt).

Typically, the desired composition of dried lipids was rehydrated with bilayer buffer (25 mM HEPES, 150 mM KCl, pH 7.4) to reach a 5 mM final concentration. The mixture was sonicated with the following protocol repeated three times cycles (5 s pulse - 5 s pause) followed by one minute pause at 4 °C. About 2 μl of each SUV solution was aspirated using neMESYS Syringe Pumps (Cetoni) into each cytosolic (top) and extracellular (bottom) channel syringe filled with bilayer buffer. After filling up the chip with squalene, the extracellular syringe was pushed with a constant flow rate at 0.005 μl/s to form an extracellular side monolayer. One minute later, the cytosolic syringe was pushed at 0.01 μl/s. Once the two electrodes at the outlet of each channel were immersed in buffer, the flow rate was decreased and kept at 0.0012 μl/s throughout the experiment.

About 1 h after the two monolayers were formed, the squalene trapped in the cylindrical hole was sufficiently absorbed to sufficiently reduce the inter-leaflet distance for bilayer nucleation at the center of the droplet. Upon additional squalene absorption, the bilayer continued zipping and expanding until it reached the inner surface of the cylindrical hole. Usually, the desired concentration of αSyn was injected at this stage at 0.01 μl/s flow rate. In specific cases (e.g., bilayer area-dependent capacitance measurement), αSyn was injected in before membrane nucleation. The bilayer formation was observed simultaneously with an EPC10 USB amplifier (HEKA) and an Eclipse Ti confocal microscope (Nikon) with add-ons, such as CSU-X spinning disk (Yokogawa), TuCam (Andor), and dual iXon Ultra camera (Andor). In general, bright-field images were recorded in 1 fps during the capacitance measurement with a 10 mV sinusoidal wave and 20 kHz frequency which was generated by the lock-in function in PatchMaster software (HEKA). When bilayer was ruptured by its native lysis tension or presence of αSyn, the capacitance showed unrealistic value.

**αSyn pore measurement.** The desired concentration of αSyn was injected at 0.01 μl/s flow rate into the cytosolic channel only or both channels (the latter was done in rare cases involving FRAP measurements). To measure ionic current across the membrane, a voltage difference from 0 mV (cytosolic channel) to 100 mV (extracellular channel) was applied to the membrane just after αSyn injection. The αSyn pore formation and enlargement were attested by step-current jumps, and the membrane rupture was measured by infinite current flow (>10 nA). Note that some current measurements showed glitches (<25 pA) regularly (1 s interval in the example presented Fig. 2b). This is due to optical recording because the glitch frequency is always the image acquisition rate and it disappears when there is no image acquisition.

**Fluorescence recovery after photobleaching.** FRAP experiments were performed with a laser scanning confocal microscope, Leica-SP5. Imaging area, 150 μm × 150 μm, was recorded as 512 × 512 pixels at 0.67 s/frames. Before and after bleaching, the area was illuminated with a 488 nm argon laser at 5% power. During the one-frame bleaching time, laser power was increased to 100% in the region of interest. Emitted light from 520 to 600 nm was collected during the whole process.

**Statistics and reproducibility.** Each experiment presented has been performed at least three times with a new chip and new lipids. Two separate batches of αSyn (catalog # S7820 from Merck) were used to perform this study. Statistics for the current jumps were obtained on 23, 24, 35, 27, 22, 30, 25 and 7 events for step number 1, 2, 3, 4, 5, 6, 7 and 8 respectively. The significance of error bars, standard error or standard error of the mean, is indicated in each figure legend.

**Reporting summary.** Further information on research design is available in the Nature Research Reporting Summary linked to this article.

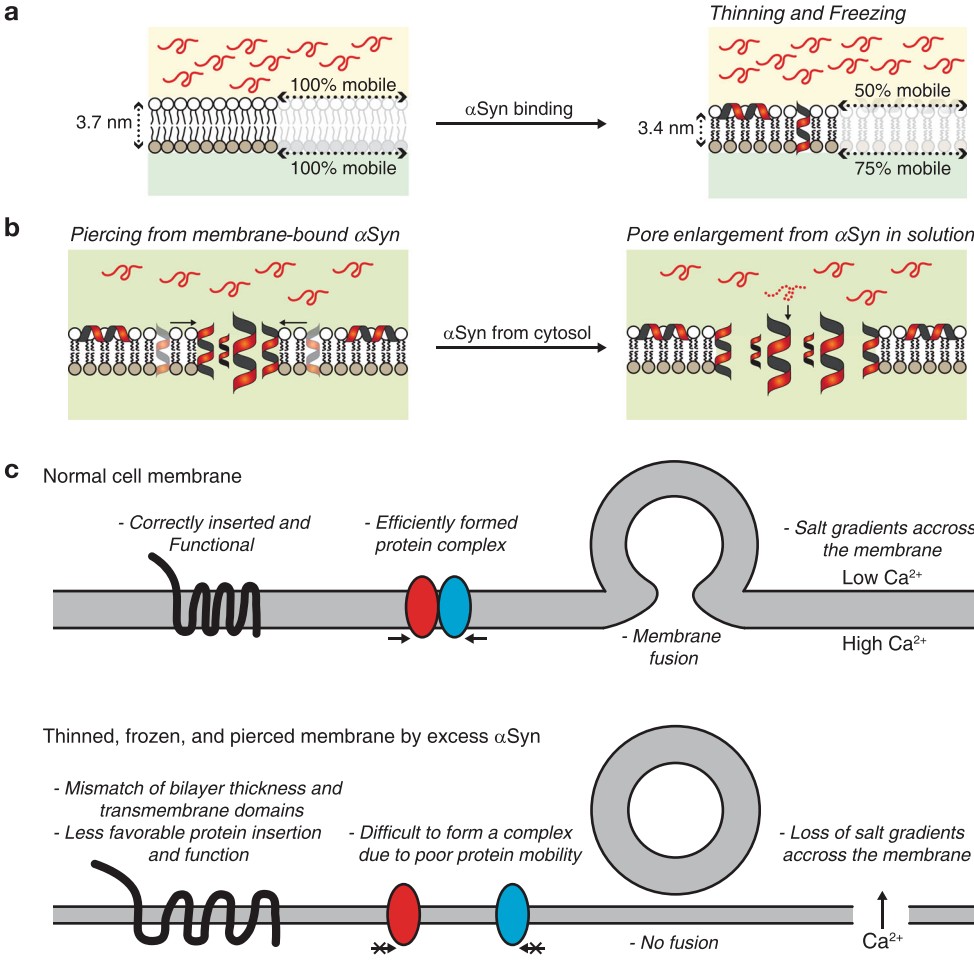

**Fig. 5 Mechanism of αSyn-mediated membrane alteration and cell malfunction. a** αSyn binds and accumulates on the cytosolic side of the membrane, leading to membrane freezing of both leaflets and capacitance change possibly due to thinning. **b** Some membrane-bound αSyn form oligomers in which 4 α-helices span the membrane, nucleating a pore. The pore is enlarged by subsequent addition of αSyn coming one by one from the cytosol. **c** Example of potential damages to neuronal activity due to αSyn-mediated simultaneous thinning, freezing and piercing of the membrane.

## Data availability

The datasets generated during and/or analyzed during the current study are available from the corresponding author on reasonable request.

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

## Acknowledgements

This work was supported by a European Research Council (ERC) funded grant under the European Union's Horizon 2020 research and innovation programme (grant agreement no. 669612) and the French National Research Agency (Grant ANR-14-1CHN-0022-01).

## Author contributions

P.H. and F.P. designed the experiments, analyzed the data and wrote the article. P.H. collected the data.

## Competing interests
The authors declare no competing interests.
