## [Peer Review File · Communications Biology]

Reviewers' comments:

Reviewer #1 (Remarks to the Author):

The manuscript by Paul and Frederick, demonstrate the influence of Alpha Synuclein aggregation on in vitro asymmetric membranes made using a custom-built microfluidic setup. Firstly, the authors show that synuclein aggregation on the asymmetric membranes results in formation of pores of discrete sizes that expand over time by assembly of unit blocks of the peptide. The induced pore formation is proved by membrane capacitance measurements. Secondly, the authors show that synuclein aggregation results in thinning of the membrane. And finally, using fluorescence recovery curves authors show that synuclein insertion immobilizes both the leaflets of the asymmetric membrane. Overall, the manuscript is very interesting focussing on a novel reconstitution approach, technically strong and well written!

However, there are a few important concerns that need to be addressed before the manuscript can be accepted for publication.

Concerns to be addressed:

1) It is not clear how an SUV (a stable bilayer structure) gets perturbed and peeled apart to release monolayers that seal to form a membrane bilayer. I also referred to the root method paper published in Small, hoping to understand the same, however, the discussion on this is lacking. Generally, free lipids in solution can form such spontaneous bilayers driven by hydrophobic forces as is also mentioned in authors method paper published in Small. However, in this the SUVs are used as the source of bilayer formation. SUVs are usually considered quite stable and it is difficult to understand how the bilayer of the SUV peels its monolayers apart while interaction with Squalene resulting in two monolayers forming a bilayer. For instance, SUVs adsorbed on a surface collapse to form a supported bilayer or flattened bi-bilayer cushions. How can one rule out the formation of a bi-bilayer under the current scenario? I could not find any reference on this particular aspect discussed in the manuscript. Some discussion on this aspect is important.

2) Authors claim that: "Alpha- synuclein aggregation process is fast enough to be completed much before the membrane rupture". While there's variation in synuclein aggregation found in literature but in general alpha-synuclein is considered a very slow aggregator (takes few hours at least) even in the presence of lipid templates, even at much higher concentrations compared to what the authors use in their study. It is therefore important to mention and discuss the synuclein preparation that authors use and if there is any presence of seeds. Further, a screening of lipid dependent ThT kinetics of alpha synuclein at similar concentrations can help gain more insights into this. Authors can be more precise in the usage of the terminology "aggregation" and instead discuss the same in terms of the phase of the synuclein morphology (i.e, whether it is an oligomeric or elongation or fibrillar phase) that triggers the membrane rupturing.

3) Authors also claim that Alpha-synuclein is inserting in the membrane forming the pores based on capacitance measurements. What makes the authors reason that it initially loads on the membranes laterally and then insert, only after which pores are generated? Aren't changes in capacitance possible by multiple leakages in the asymmetric membranes arising due to just lateral loading (i.e, as synuclein elongates inducing more defects) and not only longitudinal insertion? A more sensitive way to establish the orientation of the peptide loading laterally onto the membrane could be to test the

anisotropy/lipid order changes in a depth-dependent manner by fluorescence spectroscopy using depth sensitive dyes. This would more convincingly establish the contribution of lateral loading and insertion towards leakage.

4) Authors state that they seem to observe $<25\text{pA}$ current regularly for some measurements. Could this be simply due to transient pores formed as a result of lipid defects by peptide induced phase separation upon lateral loading or elongation of the peptide? before it takes up a more stable insertion orientation. Just curious!

5) The schematics for the FRAP experimental design/proposed model (Fig 4c, 5a) might be slightly misleading as the peptide is shown laterally aligned, while the discussion suggests freezing of the both leaflets as a result of the peptide spanning both leaflets. The authors may edit the schematic slightly to incorporate a mixture of orientations to show the same.

Reviewer #2 (Remarks to the Author):

This manuscript presents work on the effect of alpha-synuclein on the behavior of a plasma membrane model. The membrane is built in a fluidic PDMS-based chamber by zipping two lipid monolayers at approaching water-oil interfaces where the sandwiched oil (squalene) leaks away in the PDMS. The system allows direct observation of the adsorption of alpha-synuclein while measuring capacitance changes and current across the formed bilayer. The authors explore both symmetric and asymmetric bilayers and interpret the collected data as resulting from arrangement of the protein to form pores of different diameters. FRAP measurements are interpreted as bilayer "freezing" due to the protein adsorption.

The work is interesting but the employed experimental system raises questions about the observed phenomena in particular considering effects associated with presence of oil traces, membrane tension build up with time and the data interpretation.

Detailed comments:

1. The authors should first show that the membrane properties and the measurements are not affected by the preparation procedure and mainly the presence of traces of squalene. In particular, evidence should be shown that the bilayer is oil free and what one measures is occurring at this membrane and not say at the rim which is contact with the PDMS. Maybe one way to do this is to provide data for the actual membrane capacitance (not simply capacitance increase) of a simple membrane composition and compare with literature data (see e.g. DOI: 10.1039/c2sm07105c and references therein). Several data sets for bare symmetric membranes (as in Fig. 1d) should be included to show what the measurement reproducibility is and the respective calculated membrane capacitance should be indicated.

2. Regarding the effect of alpha-synuclein, instead of showing a trace where the protein was initially added, as it seems to be the case in Fig 1d, a dataset of the time dependent capacitance increase should be included where the membrane is first formed and then alpha-synuclein introduced. Again, reproducibility should be demonstrated.

3. The measured current traces for the 250 nM alpha-synuclein concentration shown in Fig 2 b and d are difficult to compare as the scale in one case is in nA and in the other in pA – it is impossible to distinguish whether the signal fluctuations are not comparable to noise. More importantly, how do we know that the signal is not resulting from some kind of interaction of the protein with the edges of the formed bilayer in the vicinity of the cylindrical hole in the chamber and with traces of squalene in the

membrane?

4. The data in Fig 3a appears largely over-interpreted. It is unclear how different steps (from 1 to 8) can be resolved from such noisy data. The figure does not really show step-like changes.
5. The interpretation of the capacitance data in terms of changes in the membrane thickness, is also questionable as significant area increase as a consequence of alpha-synuclein insertion in the membrane has been already reported (DOI: 10.1039/c4cp05883f). Indeed, the argument at the bottom of page 9 does not make much sense – area increase as a consequence of protein insertion is not related to stretching and lysing the membrane. Indeed, the area increase in this setup is probably difficult to directly detect because of the lack of control of membrane tension. In this direction, it is also unclear whether the membrane tension increases with time, leading to the bilayer rupture especially in the asymmetric membrane as affinity of lipids in one leaflet to PDMS could be different to that of the lipids in the other leaflet; this then could lead to changes in the spontaneous curvature and associated tension.
6. Changes in the relative permittivity of the membrane also shouldn't be ignored because even hydrophilic sugars have been speculated to affect the membrane capacitance (DOI: 10.1016/j.colsurfa.2018.05.011).
7. The FRAP data in Fig. 4c look very smooth. Is this real data or a schematics? Please include the data for the pre-bleaching period.
8. Regarding the bilayer "freezing" by alpha-synuclein, the results should be discussed in terms of changes in the thermodynamic state of the membrane or protein scaffolding in view of DOI: 10.1073/pnas.1601899113, which shows that phase separation and protein aggregation is specific to the membrane composition.

Minor remarks:

- The statement in the abstract for alpha-synuclein insertion that "immobilizes both leaflets" is unclear. The authors presumably mean lipid and not bilayer immobilization.
- The composition of the leaflets in the different figures should be given in the captions.
- The authors should give some justification for the choice of the membrane leaflet compositions.
- Indicate the size of the bleached spot and the acquisition speed in the text and briefly explain the FRAP data analysis (e.g. was diffusion during bleaching taken into account).

Reviewer #3 (Remarks to the Author):

The manuscript entitled 'Nanometric thinning, freezing and piercing of in vitro asymmetric plasma membrane by alpha-synuclein' by Heo and Pincet elucidates the effect of alpha-synuclein on membrane topology, morphology and dynamics using a microfluidic setup. The authors intelligently designed the microfluidic setup to control the lipid composition of cytosolic and extracellular leaflets to generate asymmetric membranes. The authors exploited multiple methodologies to justify their claims of pore formation, membrane freezing and membrane thinning, and provided a plausible mechanism of alpha-synuclein toxicity. However, there are certain questions that need to be addressed before the manuscript is being accepted.

1. Authors should mention the lipid composition for the symmetric membrane that they have used in their study.
2. If alphaSyn-induced membrane disruption is mainly controlled by the higher binding affinity of the protein to the cytosolic leaflet of the asymmetric membranes, authors should have used two sets of symmetric membranes with cytosolic and extracellular lipid compositions to evaluate the effect of alphaSyn on symmetric membranes to discriminate between lipid composition and symmetry of the membrane.

3. Authors should add the bilayer lifetime data in absence of α Syn for symmetric and asymmetric membranes in Fig 1(g), so that the readers could visualize the effect of α Syn on the bilayer lifetime.
4. It would be interesting to flow α Syn through the extracellular channel to clearly elucidate that the pore formation is being driven chemically, which is important to understand the process.
5. How do authors extrapolate the results obtained in microfluidic chamber to the cellular context? The lipid dynamics seems much slower (observed from FRAP experiment) in microfluidic setup compared to the lipid dynamics in the vesicular or cellular environment. Moreover, the excess line tension/constraint dynamics of the membrane in the microfluidic setup might be providing bias to the α Syn-induced deformation of the membrane.

Point-by-point response to reviewers

The black color and italics indicate the reviewers' original text. Our response is indicated in blue color. First of all, we wish to thank the reviewers for their positive support and for their comments that will help make the manuscript clearer.

Reviewers' comments:

Reviewer #1 (Remarks to the Author):

The manuscript by Paul and Frederick, demonstrate the influence of Alpha Synuclein aggregation on in vitro asymmetric membranes made using a custom-built microfluidic setup. Firstly, the authors show that synuclein aggregation on the asymmetric membranes results in formation of pores of discrete sizes that expand over time by assembly of unit blocks of the peptide. The induced pore formation is proved by membrane capacitance measurements. Secondly, the authors show that synuclein aggregation results in thinning of the membrane. And finally, using fluorescence recovery curves authors show that synuclein insertion immobilizes both the leaflets of the asymmetric membrane. Overall, the manuscript is very interesting focussing on a novel reconstitution approach, technically strong and well written! However, there are a few important concerns that need to be addressed before the manuscript can be accepted for publication.

Concerns to be addressed:

1) It is not clear how an SUV (a stable bilayer structure) gets perturbed and peeled apart to release monolayers that seal to form a membrane bilayer. I also referred to the root method paper published in Small, hoping to understand the same, however, the discussion on this is lacking. Generally, free lipids in solution can form such spontaneous bilayers driven by hydrophobic forces as is also mentioned in authors method paper published in Small. However, in this the SUVs are used as the source of bilayer formation. SUVs are usually considered quite stable and it is difficult to understand how the bilayer of the SUV peels its monolayers apart while interaction with Squalene resulting in two monolayers forming a bilayer. For instance, SUVs adsorbed on a surface collapse to form a supported bilayer or flattened bi-bilayer cushions. How can one rule out the formation of a bi-bilayer under the current scenario? I could not find any reference on this particular aspect discussed in the manuscript. Some discussion on this aspect is important.

This is an important technical point that is well-known in the DIB (Droplet Interface Bilayer) community but the reviewer is correct this is not obvious for a general audience. SUV tend to be disrupted when they collide with a hydrophobic/hydrophilic interface where they can form a monolayer. A typical case is that of an oil/buffer interface. This is described in the early work by Bayley's group at Oxford (see for instance Hwang et al. J. Am. Chem. Soc. **2008**, 130, 5878–5879 – Figure 1b and related discussion). This is now referred to in the manuscript.

2) Authors claim that: "Alpha- synuclein aggregation process is fast enough to be completed much before the membrane rupture". While there's variation in synuclein aggregation found in literature but in general alpha-synuclein is considered a very slow aggregator (takes few hours at least) even in the presence of lipid templates, even at much higher concentrations compared to what the authors use in

there study. It is therefore important to mention and discuss the synuclein preparation that authors use and if there is any presence of seeds. Further, a screening of lipid dependent ThT kinetics of alpha synuclein at similar concentrations can help gain more insights into this. Authors can be more precise in the usage of the terminology "aggregation" and instead discuss the same in terms of the phase of the synuclein morphology (i.e, whether it is an oligomeric or elongation or fibrillar phase) that triggers the membrane rupturing.

We agree that the term aggregation is too vague and we probably used it inappropriately.

α -synuclein quickly binds to the membrane (optically the membrane seems covered in minutes, see Figure 1 e and f). What we meant by aggregation is this global binding of α -synuclein to the membrane, not the aggregation of α -synuclein itself into oligomers and/or fibrils, that, indeed, takes hours to days. We apologize for the use of this term and changed it throughout the text to be more accurate.

Tracking ThT kinetics to observe the fibril formation on the membranes is an excellent idea but our setup, in which membranes are stable for up to 3-4 hours, is not appropriate.

3) Authors also claim that Alpha-synuclein is inserting in the membrane forming the pores based on capacitance measurements. What makes the authors reason that it initially loads on the membranes laterally and then insert, only after which pores are generated? Aren't changes in capacitance possible by multiple leakages in the asymmetric membranes arising due to just lateral loading (i.e, as synuclein elongates inducing more defects) and not only longitudinal insertion? A more sensitive way to establish the orientation of the peptide loading laterally onto the membrane could be to test the anisotropy/lipid order changes in a depth-dependent manner by fluorescence spectroscopy using depth sensitive dyes. This would more convincingly establish the contribution of lateral loading and insertion towards leakage.

The reviewer is absolutely right. We cannot differentiate lateral loading from longitudinal insertion. We have modified the manuscript by presenting a mixture of both longitudinal insertion and lateral loading to show this in all schematics. However, we can say that, if α synuclein inserts longitudinally, it does not induce leakage by itself because we would immediately see it by current measurements. At least 4 α -helices are necessary to open (probably 2 proteins each contributing for 2 α -helices, see discussion). This could absolutely be the result of 2 longitudinally inserted α synucleins that bind to each other. This is now mentioned in the last part of the results (membrane thinning) and the first paragraph of the discussion.

4) Authors state that they seem to observe <25pA current regularly for some measurements. Could this be simply due to transient pores formed as a result of lipid defects by peptide induced phase separation upon lateral loading or elongation of the peptide? before it takes up a more stable insertion orientation. Just curious!

We are certain this less than 25pA currents are due to the camera. When changing the acquisition frequency of the camera, they also change commensurably (the frequency of these glitches is exactly the acquisition rate) and when there is no simultaneous observation, they simply disappear (e.g.

compare figure 2b –with camera- and 2c –without camera). In most experiments we chose to keep the camera on to have simultaneous optical and electrical observations. We indicated this point in the Methods section (α Syn pore measurement).

5) The schematics for the FRAP experimental design/proposed model (Fig 4c, 5a) might be slightly misleading as the peptide is shown laterally aligned, while the discussion suggests freezing of the both leaflets as a result of the peptide spanning both leaflets. The authors may edit the schematic slightly to incorporate a mixture of orientations to show the same.

The reviewer is correct. This is related to point 3. We now fully consider the possibility of longitudinal insertion and we suggest that such insertion could explain the freezing of both leaflet. We have modified the figures accordingly. We thank the reviewer for this suggestion.

Reviewer #2 (Remarks to the Author):

This manuscript presents work on the effect of alpha-synuclein on the behavior of a plasma membrane model. The membrane is built in a fluidic PDMS-based chamber by zipping two lipid monolayers at approaching water-oil interfaces where the sandwiched oil (squalene) leaks away in the PDMS. The system allows direct observation of the adsorption of alpha-synuclein while measuring capacitance changes and current across the formed bilayer. The authors explore both symmetric and asymmetric bilayers and interpret the collected data as resulting from arrangement of the protein to form pores of different diameters. FRAP measurements are interpreted as bilayer “freezing” due to the protein adsorption.

The work is interesting but the employed experimental system raises questions about the observed phenomena in particular considering effects associated with presence of oil traces, membrane tension build up with time and the data interpretation.

Detailed comments:

1. The authors should first show that the membrane properties and the measurements are not affected by the preparation procedure and mainly the presence of traces of squalene. In particular, evidence should be shown that the bilayer is oil free and what one measures is occurring at this membrane and not say at the rim which is contact with the PDMS. Maybe one way to do this is to provide data for the actual membrane capacitance (not simply capacitance increase) of a simple membrane composition and compare with literature data (see e.g. DOI: 10.1039/c2sm07105c and references therein). Several data sets for bare symmetric membranes (as in Fig. 1d) should be included to show what the measurement reproducibility is and the respective calculated membrane capacitance should be indicated.

We agree with the reviewer that this point is critical. However, we have already discussed it in our previous article describing the technique (Small **2019**, 1900725 , DOI: 10.1002/smll.201900725). Our measurement of capacitance is perfectly linear with the membrane area (e.g. Fig. 3b in the previous article). We have done this for several membrane compositions and the resulting specific capacitances are in perfect agreement with the ones from the literature. The composition we characterized thoroughly is DOPC:DOPS:Cholesterol:DOPE (35:10:25:30 mol%) for which we found $0.75 \pm 0.04 \mu\text{F}/\text{cm}^2$ for the specific capacitance. We did not find any publication with exactly the same composition but, as an example, DOPC:DOPS (80:20 mol %) is $\sim 0.85 \mu\text{F}/\text{cm}^2$ and DOPC:cholesterol (80:20 mol %) is $\sim 0.8 \mu\text{F}/\text{cm}^2$ (Graten et al. PNAS, vol. 114, pp 328–333, 2017, note that both DOPS and cholesterol slightly decreases the specific capacitance of DOPC, hence our measurement is in perfect agreement with the published ones). Specific capacitance is how most groups test the purity of their painted membranes. However, we concur with the reviewer that it does not mean there is no trace of oil. Our setup has the advantage of allowing simultaneous optical observations. We can visualize single fluorescent molecules and we never saw any trace of fluorescent oil. We also detailed this in our previous article. Here is what we specifically wrote:

“There was no visible trace of fluorescence in the inner region surrounded by the ring in bright-field, indicating that the oil was entirely removed from the interspace between the two leaflets. During dilation, the fluorescence overlapped with the outer region of the ring and finally vanished when the

ring reached the edge of the hole. Notably, and contrarily to many planar bilayer setups, there is no observable amount of remaining solvent annulus in a fully extended bilayer.”. So we believe we have one of the best proof of an oil-free bilayer. But we may very well have missed an article with a similar demonstration.

We modified the manuscript to specifically mention this point and refer to our previous article (caption of Figure 1b of the new version of the manuscript).

2. Regarding the effect of alpha-synuclein, instead of showing a trace where the protein was initially added, as it seems to be the case in Fig 1d, a dataset of the time dependent capacitance increase should be included where the membrane is first formed and then alpha-synuclein introduced. Again, reproducibility should be demonstrated.

We thank the reviewer for this suggestion. From the comment, we realized that our protocol was not well described (we hope it is now better explained in Fig. 1b). During the experiment, we usually inject α -synuclein during the bilayer zipping and simultaneously switched to “current mode” to observe pore formation. To perform the experiment suggested by the reviewer, we injected α -synuclein after the capacitance plateau is reached and continuously monitored the capacitance until membrane rupture without switching to current mode. We performed this experiment three times with three different chips and we indeed always observed an increase in capacitance confirming our previous observation obtained on an average of 10 measurements (former Fig. 1d, now in Fig. 4a). We present one of them in the supplementary information (Fig. S3). Below are the results of the other two experiments. The arrow indicates injection of aSyn, the membrane breaks at the end of the red line. As a reference, the black line represents a protein-free membrane with the same asymmetric composition in the same chip (the plateau is proportional to the membrane area and therefore slightly varies with the chip).

3. The measured current traces for the 250 nM alpha-synuclein concentration shown in Fig 2 b and d are difficult to compare as the scale in one case is in nA and in the other in pA – it is impossible to distinguish whether the signal fluctuations are not comparable to noise. More importantly, how do we know that the signal is not resulting from some kind of interaction of the protein with the edges of the formed bilayer in the vicinity of the cylindrical hole in the chamber and with traces of squalene in the membrane?

We added one panel in Fig. 2d to make it comparable with 2b and 2c and kept the previous panels to better see pore formation. The pore signal is clearly out of the noise.

The second question is a general question about any suspended bilayer (e.g. painted or Montal-Mueller or other) that have been used for decades. We believe we have a strong case to support the absence of squalene in the membrane (see point 1). Moreover, we observe similar pore when the membrane is not fully zipped (i.e. when the bilayer is not in contact with the PDMS) and when the squalene is fully absorbed (we now explicitly say it at the end of the section entitled “ α Syn forms long-lasting pores in asymmetric membranes”), so pore opening is not due to the contact with PDMS. Finally, as we mention in the manuscript, we are not the first ones to observe pore formation due to α -synuclein, so it was expected that pores would form. Here, we demonstrate these pores also form on asymmetric plasma-membrane-like membranes and characterize them.

4. The data in Fig 3a appears largely over-interpreted. It is unclear how different steps (from 1 to 8) can be resolved from such noisy data. The figure does not really show step-like changes.

This is because we chose to present a broad view where many opening and closing pores occur. This representation gives a sense of the variation but the plateaus and steps appear very small and cannot be well seen. To circumvent the issue, we added four panels with zoom in to exemplify the current jumps (Supplementary Figure S2). We show jumps between steps 0 and 1, 1 and 2, and 2 and 3. The last panel shows an example of expansion and reduction of a pore between states 0, 1, 2, and 3. These are from 4 different pores to show the reproducibility of the steps. On figures 3b are reported the values of the steps. We modified the text to explain it.

5. The interpretation of the capacitance data in terms of changes in the membrane thickness, is also questionable as significant area increase as a consequence of alpha-synuclein insertion in the membrane has been already reported (DOI: 10.1039/c4cp05883f). Indeed, the argument at the bottom of page 9 does not make much sense – area increase as a consequence of protein insertion is not related to stretching and lysing the membrane. Indeed, the area increase in this setup is probably difficult to directly detect because of the lack of control of membrane tension. In this direction, it is also unclear whether the membrane tension increases with time, leading to the bilayer rupture especially in the asymmetric membrane as affinity of lipids in one leaflet to PDMS could be different to that of the lipids in the other leaflet; this then could lead to changes in the spontaneous curvature and associated tension.

We agree with the reviewer that we cannot be certain of the reason for the capacitance increase (we never claimed otherwise in the initial version of the manuscript). We also agree that we overlooked the membrane area increase associated with the thinning (to keep a constant volume). This does not qualitatively change our conclusions but reduces the thinning effect which was initially estimated at 0.5 nm down to 0.3 nm. We are grateful to the reviewer for noticing it. However, we do not fully understand the reviewer’s comment about the lysis tension. Is the reviewer suggesting that α -synuclein itself would increase the membrane area by ~20% in a few minutes by inserting inside the membrane? This would indeed be interesting but is really not reasonable. In any case, because we now discuss the area increase associated with the thinning process, we removed any reference to lysis tension. Actually

the area increase remains in the range of that occurring at the lysis tension (5-10%) which makes sense and is consistent with the facilitated opening of pores.

We also agree that, like in all suspended membranes (apart from GUVs), the surface tension is not controlled and often increases in time, which explains the eventual rupture of the membranes in these systems (and why in most cases non-physiological lipids, like DPhPC, are used). However, in the present case, membrane rupture happens ten times faster with α -synuclein than without (~15 minutes instead of 3 hours) indicating that the surface tension increase is primarily due to α -synuclein and the associated thinning/stretching process (see previous paragraph) and not to the spontaneous stretching of the membrane in the microfluidic chip (which occurs over hours).

6. Changes in the relative permittivity of the membrane also shouldn't be ignored because even hydrophilic sugars have been speculated to affect the membrane capacitance (DOI: 10.1016/j.colsurfa.2018.05.011).

The reviewer is right but in the suggested reference ~100mM sucrose is required to have a significant change. This is 10^6 times more than our typical α -synuclein concentration. Hence, it is difficult to use this article to support this assumption. Nevertheless, we now emphasize more that a change in relative permittivity cannot be excluded. We still lean towards a thinning of the membrane because this was also previously suggested in GUV experiments (as indicated in the initial version of the manuscript). The two results are in perfect agreement and make sense, this is why we prefer to put emphasis on it (without ruling out the alternate possibilities). Again, we changed the text and hope it is now clearer.

7. The FRAP data in Fig. 4c look very smooth. Is this real data or a schematic? Please include the data for the pre-bleaching period.

The reviewer is absolutely right, we forgot to include the data points. Only the fitting lines were drawn in the original figure. Figure 4c is now modified with the data points and pre-bleaching period. We apologize for the mistake and thank the reviewer for noticing it.

8. Regarding the bilayer "freezing" by alpha-synuclein, the results should be discussed in terms of changes in the thermodynamic state of the membrane or protein scaffolding in view of DOI: 10.1073/pnas.1601899113, which shows that phase separation and protein aggregation is specific to the membrane composition.

There is a confusion due to our inappropriate use of the term "aggregation". What we meant by aggregation is the binding and accumulation of α -synuclein to the membrane and not fibrils/oligomer formation (which is studied in the reference suggested by the reviewer). This is detailed in our response to point 1 of reviewer 1. We have removed any reference to "aggregation" and apologize for the confusion.

Minor remarks:

- *The statement in the abstract for α -synuclein insertion that “immobilizes both leaflets” is unclear. The authors presumably mean lipid and not bilayer immobilization.*

The reviewer is correct. We modified the abstract accordingly.

- *The composition of the leaflets in the different figures should be given in the captions.*

We have added the compositions in the captions.

- *The authors should give some justification for the choice of the membrane leaflet compositions.*

As indicated in the first paragraph of the results section, the membrane leaflet composition is as close as can be from that of the (asymmetric) plasma membrane to better mimic physiology. We have added a reference to the corresponding sentence.

- *Indicate the size of the bleached spot and the acquisition speed in the text and briefly explain the FRAP data analysis (e.g. was diffusion during bleaching taken into account).*

The disk diameter was 10 μ m. The bleach time for each pixel was less than 5 μ s. We added the information in the caption.

Reviewer #3 (Remarks to the Author):

The manuscript entitled 'Nanometric thinning, freezing and piercing of in vitro asymmetric plasma membrane by α -synuclein' by Heo and Pincet elucidates the effect of α -synuclein on membrane topology, morphology and dynamics using a microfluidic setup. The authors intelligently designed the microfluidic setup to control the lipid composition of cytosolic and extracellular leaflets to generate asymmetric membranes. The authors exploited multiple methodologies to justify their claims of pore formation, membrane freezing and membrane thinning, and provided a plausible mechanism of α -synuclein toxicity. However, there are certain questions that need to be addressed before the manuscript is being accepted.

1. Authors should mention the lipid composition for the symmetric membrane that they have used in their study.

The composition for the symmetric membrane is added in the Experimental section and in the captions of figures 1, 2 and 4.

2. If α Syn-induced membrane disruption is mainly controlled by the higher binding affinity of the protein to the cytosolic leaflet of the asymmetric membranes, authors should have used two sets of symmetric membranes with cytosolic and extracellular lipid compositions to evaluate the effect of α Syn on symmetric membranes to discriminate between lipid composition and symmetry of the membrane.

We thank the reviewer for this suggestion. As indicated in our previous article describing the technique (Small **2019**, 1900725 , DOI: 10.1002/smll.201900725 – Figure 5a) not all lipid compositions lead to long-lasting membranes. Sadly, a symmetric membrane made of the extracellular leaflet composition (DOPC:DOPS:Chol:DOPE:SM 20:5:50:15:10) is stable for only a few minutes. Instead, we have kept the asymmetric bilayer and injected α Syn in the extracellular channel. The average lifetime of the bilayer is close to that of the symmetric bilayer made of the intracellular leaflet (DOPC:DOPS:Chol:DOPE 10:15:45:30). This is now indicated in the main text and in Supplementary Figure S1a.

3. Authors should add the bilayer lifetime data in absence of α Syn for symmetric and asymmetric membranes in Fig 1(g), so that the readers could visualize the effect of α Syn on the bilayer lifetime.

We have added the lifetime without α Syn. Thank you for the suggestion.

4. It would be interesting to flow α Syn through the extracellular channel to clearly elucidate that the pore formation is being driven chemically, which is important to understand the process.

It has been addressed in our response to comment #2.

5. How do authors extrapolate the results obtained in microfluidic chamber to the cellular context? The lipid dynamics seems much slower (observed from FRAP experiment) in microfluidic setup compared to

the lipid dynamics in the vesicular or cellular environment. Moreover, the excess line tension/constraint dynamics of the membrane in the microfluidic setup might be providing bias to the α Syn-induced deformation of the membrane.

The diffusion coefficient is actually high ($12 \mu\text{m}^2/\text{s}$). This is because the recovery time depends varies linearly with the bleached area and we bleached a disk with a $10\mu\text{m}$ diameter (typically the size of a cell), which is much bigger than what is usually done In vivo. This is the reason why the recovery appears slower than in cellular environment. To avoid confusion, we now clearly indicate the bleaching area and the diffusion coefficient in the figure caption.

Line tension is only relevant at the rim of the pore and only depends on the local molecular composition. Hence, there is no reason for it to be affected by the microfluidic setup.

Reviewers' comments:

Reviewer #1 (Remarks to the Author):

The authors have addressed the raised concerns satisfactorily. The manuscript may now be accepted.

Reviewer #2 (Remarks to the Author):

The authors have addressed all my concerns and the manuscript may be acceptable.

However, it would always help the reviewer if the changes are highlighted and authors mention the page and line number in their rebuttal.

Reviewer #3 (Remarks to the Author):

In the revised manuscript of Heo et al the authors present a novel and interesting setup to measure adsorption and subsequent membrane remodeling by α -synuclein. The authors use a combination of optical and electric methods to quantitatively characterize the interaction and, ultimately, pore formation of a suspended membrane. Generally the manuscript presents a detailed and solid study which is supported by a number of quantitative measurements and as such is a strong candidate for publication in *Communication Biology*. However some methodological information seems to be missing and I have some concerns with the interpretation of the data. The concerns are summarized below:

- In line 85 the authors claim the "distribution of lifetimes does not seem to depend on concentration (Fig 1g). In Figure 1g I see not the distribution but the mean and errorbars (please define the meaning of S.D. or S.E. in the caption). Additionally the mean does seem to depend on concentration at lower values, but it is hard to say if this is significant. If the authors want to maintain this claim the actual lifetime distribution should be shown and analyzed.
- The authors deduce "discrete distribution of the current intensity" from Fig 3a and Supplementary Figure 2, this does not seem so clear to my eye. It would be good to plot a histogram of the actual current steps measured and show that distinct bins appear, see also the next point.
- The quantitative analysis in Fig 3 b,c is very nice but it remains unclear how the error-bars were calculated. Are the errorbars deduced from the uncertainty of the height of the current jump or some other estimate? Please explain.
- From the data in Figure 4a it is clear that the measured capacitance is a function of the protein concentration. But I can't follow the interpretation in terms of the thickness and area. Reading the manuscript I have assumed that the (apparent) membrane area is fixed by the geometry of the PDMS chip. Is this true or can the area change also be detected by the optical measurements? I assume not, as the authors do not mention this. Surely some "sub-optical" membrane reservoirs can exist in e.g. thermal fluctuations, but these are usually only 1-2 % of the total area and seem not large enough to explain the capacitance variations found. On the other hand the suggested change in permittivity seems to be not taken into account in the calculation, even if protein adsorption and maybe nanoscale disruption of the bilayer could induce some e.g. water channels into the bilayer. This would surely (locally) change the permittivity greatly. Given these uncertainties and the ad-hoc argument of 10% area and 10% thickness change (line 190) the data in Fig 4b seems to be unreliable and maybe the best course of action is to omit this calculation and only deduce that the capacitance data indicates some structural changes in the bilayer by α -synuclein adsorption which might be a combination of multiple mechanisms.
- What was a typical α -synuclein density on the membrane (as measured by fluorescence?)

Minor points:

- The manuscript is focused on the differences in pore formation between symmetric and asymmetric membranes but seem to fall short of an actual mechanistic explanation. Maybe it would be helpful to mention that differences in monolayer composition give rise to a rather large interfacial tension (<https://www.ncbi.nlm.nih.gov/pmc/articles/PMC4675890/>) which will destabilize the bilayer, especially after the pore is nucleated.
- I didn't find the methodical details on the setup for the capacitance / current measurements.

Point by point response to Reviewer #3's comments

The Reviewer's comments are in italics and our response is in blue.

In the revised manuscript of Heo et al the authors present a novel and interesting setup to measure adsorption and subsequent membrane remodeling by α -synuclein. The authors use a combination of optical and electric methods to quantitatively characterize the interaction and, ultimately, pore formation of a suspended membrane. Generally the manuscript presents a detailed and solid study which is supported by a number of quantitative measurements and as such is a strong candidate for publication in Communication Biology. However some methodological information seems to be missing and I have some concerns with the interpretation of the data. The concerns are summarized below:

1. In line 85 the authors claim the “distribution of lifetimes does not seem to depend on concentration (Fig 1g). In Figure 1g I see not the distribution but the mean and errorbars (please define the meaning of S.D. or S.E. in the caption). Additionally the mean does seem to depend on concentration at lower values, but it is hard to say if this is significant. If the authors want to maintain this claim the actual lifetime distribution should be shown and analyzed.

The term “distribution” is inappropriate and we apologize for using it. We meant the mean lifetimes. We changed the text accordingly. For symmetric membranes, it is clear that the mean lifetime decreases with the concentration (red circles in Fig. 1g). For asymmetric membrane (black square), there is no apparent variation of the mean lifetime with α Syn concentration. Below is an enlarge plot. 15 minutes is within the error bar of all data points making it impossible to find any significant trend in the variation in contrast with what is observed for the symmetric membrane.

We did not see where S.D. and S.E. are indicated, so we understand the reviewer is requesting to indicate what the error bars are in each case (standard deviation, S.D., or standard error of the mean, S.E.). The reviewer is correct, we should have provided these details. We have now indicated what all error bars represent.

2. The authors deduce “discrete distribution of the current intensity” from Fig 3a and Supplementary Figure 2, this does not seem so clear to my eye. It would be good to plot a histogram of the actual current steps measured and show that distinct bins appear, see also the next point.

This question is similar to point 2 of reviewer 2 in the previous round.

Making a complete histogram of the current plateaus is difficult for two main reasons:

1. General reason: it is always a challenge to make a single histogram where 8 peaks clearly appear.
2. The current intensity steps vary as n^2 , making it impossible to choose a width for the bins.

One point the reviewer is omitting is that we actually need to follow the complete trace and count the jumps for each experiment. This allows us to obtain the n of each current plateau. Then, instead of plotting the current plateau value that varies as n^2 , we can plot the corresponding pore diameter that varies as n , making it possible to choose a bin size over the eight steps. The result is presented below and has been added in Supplementary Figure 2 (panel e).

This is the histogram of the number of observed pore vs. the diameter of the pore. For each individual pore, the diameter was obtained by averaging up to 10 points following the corresponding current jump. Each color indicates the number of steps that have been counted by following the whole current trace.

Also note that, on two occasions, we saw a current plateau for $n=9$. The mean of their diameter follows the same linear trends with n . However, because of the low number of events, we chose not to present it.

3. The quantitative analysis in Fig 3 b,c is very nice but it remains unclear how the error-bars were calculated. Are the errorbars deduced from the uncertainty of the height of the current jump or some other estimate? Please explain.

The error bars are the standard deviations on the current plateaus of the corresponding n . This is now indicated in the figure caption.

4. From the data in Figure 4a it is clear that the measured capacitance is a function of the protein concentration. But I can't follow the interpretation in terms of the thickness and area. Reading the manuscript I have assumed that the (apparent) membrane area is fixed by the geometry of the PDMS chip. Is this true or can the area change also be detected by the optical measurements? I assume not, as the authors do not mention this. Surely some "sub-optical" membrane reservoirs can exist in e.g. thermal fluctuations, but these are usually only 1-2 % of the total area and seem not large enough to explain the capacitance variations found. On the other hand the suggested change in permittivity seems to be not taken into account in the calculation, even if protein adsorption and maybe nanoscale disruption of the bilayer could

induce some e.g. water channels into the bilayer. This would surely (locally) change the permittivity greatly. Given these uncertainties and the ad-hoc argument of 10% area and 10% thickness change (line 190) the data in Fig 4b seems to be unreliable and maybe the best course of action is to omit this calculation and only deduce that the capacitance data indicates some structural changes in the bilayer by α -synuclein adsorption which might be a combination of multiple mechanisms.

Again this is similar to a previous point of reviewer 2 (point 5).

We believe we were careful enough in the text in telling the thinning effect is a possible but not fully proven explanation. As a reminder, we mentioned:

“We cannot fully exclude the possibility that the increase in capacitance is due to a change in the relative permittivity ϵ_r . However, since thinning/stretching of the membrane by α Syn binding was previously reported, we favor this explanation.”

We do not wish to change this text because we think we are cautious enough in our formulation.

The reviewer is correct that the cross section of the hole is fixed. However, the membrane is not always perfectly horizontal and 10% increase is easily achieved by a small tilt, that we cannot detect in the course of the experiment (the membrane is so fragile when α Syn is bound that it will break as soon as we attempt to make a z-stack).

5. What was a typical α -synuclein density on the membrane (as measured by fluorescence?)

This is a good question that we also asked ourselves. However, we did not manage to obtain an accurate measurement so we decided not to discuss it in the manuscript. We just know that the concentration is low in the sense that there are probably between 200 and 20,000 α Syn molecules bound to the membrane at rupture, *i.e.* 1 α Syn every 10^6 to 10^8 lipids. We obtained this estimate by measuring the fluorescence in the hole above the membrane at 25 nM ($\sim 60,000$ a.u.) and the membrane intensity after washing off the unbound α Syn (~ 100 a.u.). These values were obtained with the same acquisition conditions (40x objective, 1.1 N.A., 1 sec acquisition time, same 488 nm laser excitation intensity, same camera settings; values are given after subtracting the background). The optical section is about $1\mu\text{m}$. Hence, for 25 nM, in a projected area of $1\mu\text{m}^2$ there is $25\text{ nM} \times 10^{-15}\text{ liter} = 25 \cdot 10^{-24}\text{ M} \sim 10$ molecules. This means 60,000 intensity corresponds to 10 molecules per projected μm^2 . Hence 100 intensity corresponds to 1 molecule per $60\mu\text{m}^2$. This is probably underestimated because there are several causes of errors: i) more molecules than the optical section are probably included in the 3D signal, ii) there may be quenching if oligomers form on the membrane and iii) the background (~ 50) is significant compared to the actual fluorescent signal of α Syn on the membrane (~ 100). Also, we do not see single α Syn on the fluorescent image, suggesting molecule concentration is too high for the individual dyes to be optically separated. Hence, 1 α Syn per $60\mu\text{m}^2$ is probably a lower bound, ~ 1 molecule per few μm^2 is a more likely value. We doubt that we underestimate the total amount by more than 2 orders of magnitude, which leads to the values indicated above: between 200 and 20,000 α Syn are bound to the membrane.

Minor points:

- *The manuscript is focused on the differences in pore formation between symmetric and asymmetric membranes but seem to fall short of an actual mechanistic explanation. Maybe it would be helpful to mention that differences in monolayer composition give rise to a rather large interfacial tension (<https://www.ncbi.nlm.nih.gov/pmc/articles/PMC4675890/>) which will destabilize the bilayer, especially after the pore is nucleated.*

Micrometer size lipid domains that are discussed in the reference suggested by the reviewer are probably not related to the asymmetric membrane we are currently using. On the contrary, we found that asymmetric membranes are in general more stable than the symmetric ones (as indicated in reference 32). However, the reviewer is correct, as soon as α Syn is added, the asymmetric membrane is much less stable (Fig. 1g).

- *I didn't find the methodical details on the setup for the capacitance / current measurements.*

The methodological details of capacitance measurements were at the end of the section “*Simultaneous optical and electrical monitoring of the asymmetric plasma membrane formation.*”

“In general, bright-field images were recorded in 1 fps during the capacitance measurement with a 10 mV sinusoidal wave and 20 kHz frequency which was generated by the lock-in function in PatchMaster software (HEKA).”

The methodological details of capacitance measurements were at the beginning of the section “ *α Syn pore measurement.*”

“To measure ionic current across the membrane, a voltage difference from 0 mV (cytosolic channel) to 100 mV (extracellular channel) was applied to the membrane just after α Syn injection.”

Reviewers' comments:

Reviewer #4 (Remarks to the Author):

The authors have addressed most of my comments and I think the analysis of now shown in S2e make the claim of current steps much clearer.

Unfortunately I still can't agree with the authors answer to point 4 concerning the analysis of the capacitance changes with protein coverage.

In fact the numbers given by the authors for the apparent thickness change seem widely inconsistent with the numbers of α -synuclein density on the membrane (point 5). The authors claim in Figure 4b a change of (average) thickness of about 10%. Because of the membrane fluidity, a single or cluster of α -synucleins can only change the membrane thickness in its immediate surrounding of maybe 2-3 lipid shells. If the upper limit of 1 α -synuclein per / μm^2 is true (even with a couple of orders of magnitude margin) and a 5nm^2 footprint of α -synuclein on the membrane only about a fraction of $5\text{E}-6$ of the membrane is covered. It seems impossible how this should decrease the average membrane thickness by 10% as claimed.

The authors wrote that the measure the capacitance by "capacitance measurement with a 10 mV sinusoidal wave and 20 kHz frequency which was generated by the lock-in function in PatchMaster software (HEKA)."

Does this mean that this essentially extracts the imaginary part of the total complex impedance? Maybe the leakage currents through the membrane contribute to the (apparent) capacitance measured?

Given these uncertainties I still think it is best to show the -certainly interesting and reliable - measurements of capacitance and not extract a membrane thickness in Fig 4b.

Given that it does not seem that authors agree with this approach I ask the editor to decide on this issue.

Otherwise I think the paper is good and ready to be published as is.

Point by point response to the Reviewer's comments

The Reviewer's comments are in italics and our response is in blue.

The authors have addressed most of my comments and I think the analysis of now shown in S2e make the claim of current steps much clearer.

Unfortunately I still can't agree with the authors answer to point 4 concerning the analysis of the capacitance changes with protein coverage.

- 1. In fact the numbers given by the authors for the apparent thickness change seem widely inconsistent with the numbers of α -synuclein density on the membrane (point 5). The authors claim in Figure 4b a change of (average) thickness of about 10%. Because of the membrane fluidity, a single or cluster of α -synucleins can only change the membrane thickness in its immediate surrounding of maybe 2-3 lipid shells. If the upper limit of 1 α -synuclein per / μm^2 is true (even with a couple of orders of magnitude margin) and a 5nm^2 footprint of α -synuclein on the membrane only about a fraction of $5\text{E}-6$ of the membrane is covered. It seems impossible how this should decrease the average membrane thickness by 10% as claimed.*

We are turning in circle as the answer to this comment is directly connected to our initial answer to reviewer 2 to whom we indicated that the membrane thinning and expansion cannot be due to the mere insertion of α -Syn at low density which, as the reviewer correctly calculated, cannot explain these changes. Our interpretation, which was written in the very first version of this manuscript but removed after the first round of reviews, is that the proteins induce a global increase in membrane tension that increases the total area and decreases the thickness.

From what is written in the review, it appears the reviewer concurs that the increase in capacitance can only be due to membrane thinning/expansion and/or relative permittivity increase (see eq. 2). The reviewer favors a change in permittivity and we favor a change in thickness/area. Of course, as we did from the beginning, we remain open to both explanations. We have always tried to present the experimental facts (capacitance increase) and be cautious about the interpretation. We were probably too much showing our preferred explanation (e.g., it appeared in the title). However, we are a little uncomfortable of opening too much the door on the relative permittivity because nobody ever saw any increase in relative permittivity with α Syn while area increase associated with membrane thinning was directly observed previously (on GUVs with symmetric composition). Also, we cannot physically or chemically easily explain the increase of a relative permittivity (it cannot be due to the pore opening, see next point), while the explanation for membrane thinning is straightforward. In any case, we agree better balance the two possibilities by putting more emphasis on relative permittivity and less on membrane thinning/expansion. We removed any mention to thinning in the title and present it in the abstract and text in parallel with an increase in relative permittivity, both explanations being non-mutually exclusive. We removed Fig. 4b. We hope this compromise will satisfy the reviewer.

2. *The authors wrote that they measure the capacitance by "capacitance measurement with a 10 mV sinusoidal wave and 20 kHz frequency which was generated by the lock-in function in PatchMaster software (HEKA)." Does this mean that this essentially extracts the imaginary part of the total complex impedance? Maybe the leakage currents through the membrane contribute to the (apparent) capacitance measured?*

The reviewer is correct, the capacitance is essentially the imaginary part of the total complex impedance. We do not observe any significant real part (mainly below 1 nS) until a pore opens. Hence, pore opening cannot explain the increase in capacitance. Below are traces of the conductance and capacitance with α Syn at $2\mu\text{M}$ and symmetrical bilayer (cytosolic leaflet composition).

3. *Given these uncertainties I still think it is best to show the -certainly interesting and reliable - measurements of capacitance and not extract a membrane thickness in Fig 4b.*

As mentioned in our response to point 1, we removed Fig. 4b.

4. *Given that it does not seem that authors agree with this approach I ask the editor to decide on this issue. Otherwise I think the paper is good and ready to be published as is.*

We completely agree with the reviewer that, at this point, a final editorial decision needs to be taken.

REVIEWERS' COMMENTS:

Reviewer #5 (Remarks to the Author):

I agree with the primary concern of the reviewer 2 with regard to the cause of the observed increase in capacitance (i.e, contribution of membrane thickness and relative permittivity towards the observed change in capacitance). One way of establishing correlation between the protein insertion and change in membrane thickness would require estimating the protein density of the bound synuclein (which the current paper did not estimate) as well as monitoring the changes in area of the membrane (direct measurement of which is not possible to estimate in the current set up). This technical limitation of the set up was also acknowledged by the reviewer himself.

So, considering the fact that most of the concerns were addressed by the author and given that authors have removed the figure 4b. on the change in membrane thickness as well as edited their discussion as advised by the reviewer, I think the manuscript can now be accepted.